# Energy-Efficient Collision Avoidance MAC Protocols for Underwater Sensor Networks: Survey and Challenges

Faisal Abdulaziz Alfouzan 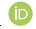

Department of Forensic Sciences, College of Criminal Justice, Naif Arab University for Security Sciences (NAUSS), Riyadh 14812, Saudi Arabia; falfouzan@nauss.edu.sa; Tel.: +966-(0)-11-246-3444 (ext. 1515)

**Abstract:** The Medium Access Control (MAC) layer protocol is the most important part of any network, and is considered to be a fundamental protocol that aids in enhancing the performance of networks and communications. However, the MAC protocol's design for underwater sensor networks (UWSNs) has introduced various challenges. This is due to long underwater acoustic propagation delay, high mobility, low available bandwidth, and high error probability. These unique acoustic channel characteristics make contention-based MAC protocols significantly more expensive than other protocol contentions. Therefore, re-transmission and collisions should effectively be managed at the MAC layer to decrease the energy cost and to enhance the network's throughput. Consequently, handshake-based and random access-based MAC protocols do not perform as efficiently as their achieved performance in terrestrial networks. To tackle this complicated problem, this paper surveys the current collision-free MAC protocols proposed in the literature for UWSNs. We first review the unique characteristic of underwater sensor networks and its negative impact on the MAC layer. It is then followed by a discussion about the problem definition, challenges, and features associated with the design of MAC protocols in UWANs. Afterwards, currently available collision-free MAC design strategies in UWSNs are classified and investigated. The advantages and disadvantages of each design strategy along with the recent advances are then presented. Finally, we present a qualitative comparison of these strategies and also discuss some possible future directions.

**Keywords:** underwater sensor networks; medium access control; contention-free MAC protocols

## 1. Introduction

Recently, underwater sensor networks (UWSNs) have been considered to be a powerful technology to observe and explore lakes, rivers, seas, and oceans. Water covers approximately two-thirds of the Earth's surface, but just a small part of it has been explored [1–6]. Therefore, UWSNs are a valuable research direction in approaching underwater applications. Due to their wide range of applications in many fields, such as environmental and pollution monitoring, oceanographic data collection, ocean samples, early warning systems, disaster prevention, offshore exploration, distributed tactical surveillance, assisted navigation, and resource discovery, UWSNs have increasingly attracted considerable attention over the last two decades [7–10]. This is aimed to improve ocean exploration and support the demand for various time-critical civilian and military aquatic applications. Aquatic applications are considered the major objective with regards to resource dedication, with the objective of decreasing the reliance on land resources. However, it would be expensive and challenging to evaluate the underwater aquatic environment [11–13].

UWSNs refer to a set of ad hoc networks that are identical to various kinds of sensor networks. UWSNs comprise numerous sensors that are dispersed underwater to conduct mutual monitoring tasks within a predetermined area [14–17]. The sensor nodes are applied in the selected UWSN applications, based on diverse demands, which are either fixed or mobility, or can be a hybrid of both states [18,19]. Furthermore, they use acoustic signals to communicate with other nodes. However, underwater sensor nodes are typically

very expensive, and are usually used in very large areas of the ocean environment, leading to a sparse deployment of networks involving mobile sensors [11].

Underwater environments have certain physical restrictions and distinctive features, which should be considered during the development of Medium Access Control (MAC) protocols. These include slow propagation delay, low available bandwidth, energy limitations, sensors movements in water current, and high deployment expenses. In addition, UWSNs have different characteristics in comparison with the terrestrial sensor networks, where UWSNs use acoustic signals that lead to reduce network performance and more extensive range within underwater environments, which is unlike both optical and electromagnetic waves [20,21]. Radio frequency (RF) waves are influenced by high attenuation in the aquatic environment, especially when the frequency is high. Consequently, high transmission power and large antennae are needed [1,21]. Optical waves are rapidly impaired by scattering and absorption in water [21–23]. Hence, a sensor node in water uses acoustic waves to communicate, which is five orders of magnitude less than that of radio waves. As a consequence of their lower propagation speed, higher propagation delays conduct in communication, even between nodes located close to each others.

The MAC protocol is generally designed to deal with the effective management of channel communications, and this objective can be achieved by sharing a medium with other sensors to prevent retransmissions and collisions on the network. Meanwhile, it contributes to support consistent network transmissions by resolving any conflict between network nodes during communication. Thus, the MAC protocol is able to provide high energy efficiency and throughput, reduced delay during communications, and fairness between nodes in the network. The MAC protocol can be categorised into two major classes, namely the contention-free and contention-based protocols [21,24–26].

Due to some inherent characteristics of underwater acoustic channels, such as high latency, limited bandwidth, and a high bit error rate, resulting in the contention-based MAC protocols are totally expensive in UWSNs. More specifically, because of the above-mentioned characteristics, the delay in packet transmission becomes very high, and hence the possibility of collisions in the random access-based MAC protocols highly increases. In the handshake-based MAC protocols, the performance significantly decreases due to the control packets exchanged (e.g., Request-To-Send/Clear-To-Send (RTS/CTS)) during the operational process which becomes an expensive task. Using the control packets in the handshake-based MAC protocols, the probability of collisions is therefore decreased, and becomes less than that in the random access-based MAC protocols. Because of the recent observations, the contentions by exchanging control packets are significantly more expensive, and thus both handshake-based and random access-based MAC cannot efficiently be operated, which are not as efficient as they are in the terrestrial networks [27–29].

Since the contention-based MAC protocols are expensive in UWSNs, the collision-free MAC protocols guarantee to achieve a high performance (i.e., improving the energy efficiency, throughput, and fairness) [12,30,31]. In this category, contention-free, communication channels are separated into frequency, code domains, and time such as Frequency Division Multiple Access (FDMA), Code Division Multiple Access (CDMA), and Time Division Multiple Access (TDMA) [17,32].

The current used MAC solutions are mostly focused on Time-division multiple access (TDMA). This is due to the incompatibility between Frequency-division multiple access (FDMA) and UWSNs, as FDMA is considered to have narrow bandwidth in acoustic channels, and also experience diffusion of band systems due to multipath channels and fading. Additionally, CDMA is considered a more effective option for frequency selective fading, which occurs because of multiple paths. Thus, CDMA is not an effective solution for UWSNs, as it also faces the challenge of addressing the near-far problem [33].

TDMA is one of the best technique that can be used properly in UWSNs. This due to its ability by sharing the frequency channel (i.e., dividing signal into multiple time slots, called duty cycling mechanism). In addition, TDMA is also able to keep reliable transmission schedules by operating an extra updating and scheduling phases in order

to keep all the nodes synchronised. It also permits nodes, those are located outside of other nodes' transmission ranges, to send packets at the same time with no chance of collision. Consequently, TDMA expands channel reuse (i.e., concurrently sending in several neighbourhoods) and to also avoid packet re-transmission, which leads to increase the network throughput and decrease the energy usage.

To the best of our knowledge, this paper is the first to investigate the design strategies of contention-free MAC protocols, and how these protocols overcome the peculiar features of underwater acoustic channels. To achieve this, we first mention the characteristics of UWSNs in Section 2. Then, in Section 3, we investigate the MAC problem and challenges in three-dimensional (3D) UWSNs. A description of duty cycling mechanism is then presented in Section 4. In Section 5, we propose a classification for all the design strategies of the collision-free MAC protocols in UWSNs. In Section 6, these design strategies qualitatively are compared in terms of their performance regarding the MAC problems and challenges. Finally, in Section 7, we conclude the paper followed by identifying some directions and guidance for the future research on the MAC techniques in UWSNs.

## 2. Characteristics of Underwater Sensor Networks

This section presents the unique characteristics of UWSNs. Underwater sensor networks pose more severe situations for MAC protocols to cope with. Due to the characteristics explained below, terrestrial MAC designs are quite impractical and cannot be employed directly in the underwater environment. Suitable MAC designs for underwater acoustic communications must therefore be developed, taking all the relevant characteristics into account.

### 2.1. Node Movement in Three-Dimensionality Area

Since UWSNs are distributed in three-dimensions (3D), unlike terrestrial networks, sensor nodes move with the water current. 3D mobile nodes in medium access control can make difficulties for allowing sensor nodes to access the communication channel. The impact of hidden terminal problems in 3D UWSNs is more heavier than it is in terrestrial networks. This is mainly because of more existing neighbouring nodes whose located in different directions. Furthermore, the topology continuously changes along with the nodes' movement [34]. The speed of the sensor node depends on the water velocity, which differs over time.

### 2.2. Impact of High Propagation Delays on MAC Protocols

The high propagation delay of underwater acoustic communications has made the contention-based MAC protocols (handshaking-based and random access-based) inefficiently perform for UWSNs, while they achieve a high performance in terrestrial networks. In this network, the propagation delay between sensors has strict requirements. For instance, sensors without applying multi transmission techniques (e.g., TDMA and FDMA), packets transmitted concurrently from several sender nodes to one destination will collide due to the propagation time of these packets is negligible compared to their transmission times. In UWSNs, however, packets transmitted at the same time by multiple sender nodes at one receiver node may arrive at different times, taking into account the non-negligible propagation time difference.

An example of two sender nodes is illustrated in Figure 1. In this figure and by considering a time-slotted approach, sensors $B$ and $C$ transmit their data packets to sensor $A$ as a one-hop neighbourhood at the beginning of a time slot. The distance between sensors $B$ and $A$ ($d_{B-A}$) is not equal to the distance between $C$ and $A$ sensors ($d_{C-A}$). Let us assume that the length of the packet is $L$ bits and the bit rate (data transmission rate) of acoustic modem is $\eta$ bits per second (bps). If these two distances, (($d_{B-A}$) and ($d_{C-A}$)), meet the inequality, as shown in the following equation, both packets transmitted by sensors $B$ and $C$ simultaneously can successfully be received by sensor $A$ with no collision [35].

$$\triangle_d =\mid d_{B-A} - d_{C-A} \mid=\mid (tp_{B-A} - tp_{C-A})\, v \mid\geq \frac{L}{\eta}v, \tag{1}$$

where $tp_{B-A}$ and $tp_{C-A}$ are the propagation delay of their transmitted packets, sensors $B$ and $C$, respectively. $v$ denotes the speed of sound in water, which is approximately 1500 m per second. By considering the typical underwater sensor network parameters, $L$ = 1024 bits and $\eta$ = 10,000 bits per second, the distance difference can be given by ($\frac{L}{\eta}v$) for simultaneous transmission, which is equal to 153.6 m. This means that the high propagation delay can be leveraged to enhance and improve the network throughput and the channel use (i.e., transmission data packets simultaneously in different neighbourhoods). In other words, the long propagation delay allows more than one sender nodes in a time slot to increase the performance of the network (i.e., system throughput and channel use).

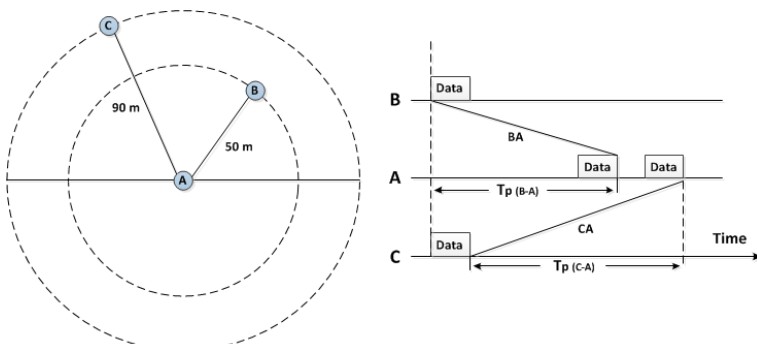

**Figure 1.** An example of two sender nodes (*B* and *C*) simultaneously send packets to node *A* in the same time slot and their transmitted packets arrive at node *A* with no collision due to the distance differences of sender-receiver pairs.

### 2.3. Impacts of Low Available Bandwidth on MAC Protocols

Underwater acoustic communications have significantly been affected by high and variable propagation delays, noise, path loss, and Doppler spread (i.e., time difference fading because of the movement of a scatter or the movement of the sender and receiver or both together leading to Doppler spread). These challenges are able to determine the spatial and temporal variability of the medium channel and make the bandwidth of the underwater acoustic communications low. The available bandwidth of the underwater acoustic channel is therefore dependent on the range and the frequency [1]. Essentially, underwater acoustic modems perform at frequencies from a few kHz to hundreds of kHz. If the underwater network operates over some tens of kilometres in a long-range system, the available bandwidth would only be a few kHz. However, for a short-range system, operating in several tens of meters may have more than a hundred kHz of bandwidth [36]. In either ways, these factors result in a low bit rate. To improve the bandwidth of the underwater acoustic channel, an accurate design of MAC protocols is required for use in UWSNs [28].

### 2.4. Acoustic Noises

Two major types of noise are considered to have a considerable impact on underwater acoustic communications; these include man-made noise and ambient noise [1,7,37]. Man-made noise is mainly caused by human activities, such as the use of pumps or shipping. Ambient noise is generated by natural occurrences such as tides and earthquakes [22,38]. The sources of these noises are shipping, power plants, and turbulence [39]. The various sources of noise cause a lossy and noisy underwater environment, which should be considered during design planning for the MAC protocols.

*2.5. Path Loss*

The aquatic environment has a higher probability of path loss than that of terrestrial physical layer. This phenomenon is caused by the attenuation, Doppler spreading, or geometric spreading [40]. The path loss can be approximated using Thorp's model [39,41,42]. More specifically, the Thorp's model is used to design the propagation of underwater acoustic communications, and to adjust the transmission power [41,43–46]. A lossy channel and the bit error rate depend on the signal frequency and traversed distance. The lossy channel or attenuation over distance $d$ with the signal frequency $f$ is defined as follows [39]:

$$A(d, f) = A_0 \, d^k \alpha(f)^d, \tag{2}$$

where $A_0$ indicates a unit-normalizing constant and $k$ is the geometric spreading factor, which sets to be 1.5 for practical scenarios. Moreover, the absorption coefficient $\alpha(f)$ is determined by the Thorp's formula. The ratio of the signal power, which includes significant data, to the unwanted signal power (noise) is known as the signal-to-noise ratio (SNR). Taking into consideration the attenuation formula, the SNR over distance $d$ with the signal frequency $f$ can be given by [39]:

$$SNR(d, f) = \frac{PR(f)}{A(d, f)PN(f)}, \tag{3}$$

where $PN(f)$ denotes the underwater environment noise and $PR(f)$ indicates the transmission power of the forwarding node with frequency $f$. To obtain the received signal without failure, the $SNR$ at the destination should be greater than a detection threshold. The ambient noise in the aquatic environment contains four essential components of turbulence $PN_t(f)$, shipping $PN_s(f)$, waves $PN_w(f)$, and thermal energy $PN_{th}(f)$, which can be calculated as [43]:

$$PN(f) = PN_t(f) + PN_s(f) + PN_w(f) + PN_{th}(f). \tag{4}$$

These different noises are dominant in the various frequency regions in which can be affected the throughput of communication channel.

*2.6. High Energy Cost*

The high energy cost is caused by high and variable propagation delays, which is another major concern in underwater sensor networks since it is difficult to replace or recharge the batteries of underwater nodes. In underwater acoustic networks, a sensor consumes much more energy than what is consumed by the terrestrial nodes (i.e., the speed of underwater acoustic communication is five orders of magnitude lower than that of terrestrial radio communication) [47,48]. Hence, energy efficiency is a fundamental requirement of MAC protocol designs in UWSNs [49].

## 3. Problem Definition and Challenges

During the design phase for resource-sharing methods in underwater acoustic networks, it is essential to consider the specific features of the channel, such those discussed in Section 2. Furthermore, considering that sending mode often expend more energy than receiving mode, therefore; re-transmission and collision should be prevented in order to reduce energy usage and enhance network's fairness and throughput. These limitations have a major impact on the design of MAC protocols, mainly because of the challenges described as follows.

*3.1. Hidden Terminal Problem*

A hidden terminal problem refers to a network node that is unaware of other nodes. Continuous communications between these nodes could result in a collision at the destination node [50,51]. Figure 2 has been shown how the hidden terminal problem can be

occurred. In this figure, when node *A* is able to see node *B* and node *C*, but both of these nodes *B* and *C* are not able to see each other. Accordingly, sending packets from both *B* and *C* nodes could cause a collision in node *A*.

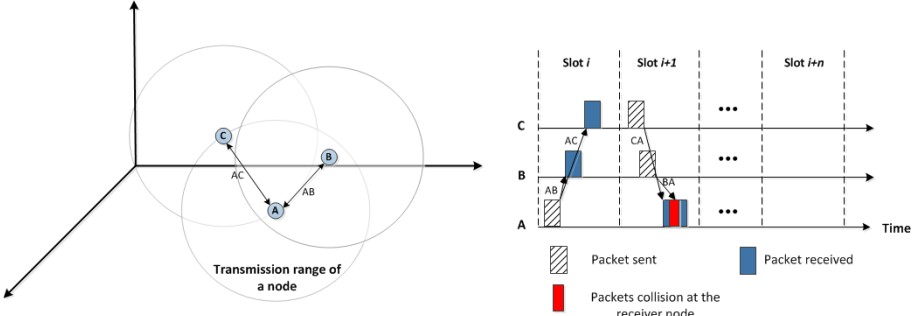

**Figure 2.** An example of hidden terminal problems.

### 3.2. Exposed Terminal Problem

The exposed terminal problem happens when a sensor is prevented by sending packets to its neighbouring sensors because it overhears with another transmission [36,52]. This issue can be seen in Figure 3, where sensor *B* and *C* are within the range of sensor *A*, but sensor *D* is not. Similarly, sensor *A* and sensor *D* are within sensor *B*'s range, but sensor *C* is not. In this case, if sensor *A* needs to transmit a packet to sensor *C*, and concurrently sensor *B* seeks to send a packet to sensor *D*, both of these transmissions could occur at the same time, because of the receivers are located out of each sensor's transmission range. Hence, sensor *A* begins sending its own packet to sensor *C*, eventually sensor *B* overhears the transmission from sensor *A* and stop its transmission to sensor *D*, because sensor *B* supposes that if it continues to send its own packet, a collision will occur.

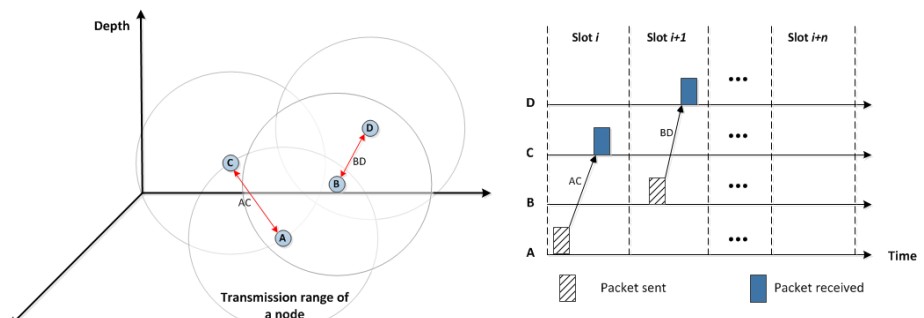

**Figure 3.** An example of exposed terminal problems.

### 3.3. Spatial-Temporal Uncertainty Problem

This problem is caused by the large propagation delay of acoustic medium. It is, therefore, essential to know the location of the sensor as well as the sending time too. This problem is known as a two-dimensional uncertainty, and it is described as follows.

- If a collision occurred at the receiver node, it would be depending on the sending time and the propagation delay between the sender and the receiver sensors. This is known as a duality that differs between the sending time and the location of the sensor nodes.
- Different distances among the underwater sensor nodes leads to uncertainty based on the current channel status, and a packet collision can be occurred even if there is no other node transmitting at the same time.

Because of the long propagation delay which may cause a collision in the aquatic network, two different example will be explained in Figure 4. The first example is shown in the right hand side, where sensors *B* and *C* start sending their own packets at the same

time. Therefore, the reception time of these two packets will be different as the propagation delays of sensors *B* and *C* are different [11,53]. The second example, as shown in the left hand side, illustrates that when both sensors *B* and *C* start sending their packets with a different times, a collision may occur at the receiver sensor node *A*.

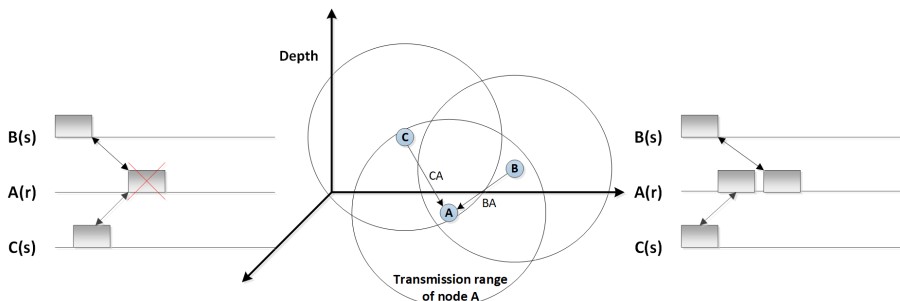

**Figure 4.** How long propagation delay highly affect the underwater MAC protocol designs.

### 3.4. Near and Far Problem

The near and far effect is a serious design challenge for MAC protocols due to the unique characteristics of underwater acoustic channels [1,4,17]. This problem can be defined as multiple sender nodes located in different distances from a receiver node to transmit their packets. Hence, the received power for all sender nodes are not almost similar, signals from distant nodes cannot be received successfully.

This problem can be shown in Figure 5, where the distance between *A* and *C* is significantly longer than the distance between *A* and *B*. As a consequence, the destination node *A* receives different signal-to-noise ratio (SNR) levels of signals originating from each of the sender nodes, due to the high level of noise produced by sensor *B*'s signals. In this case, the transmission power of each sender node should be controlled.

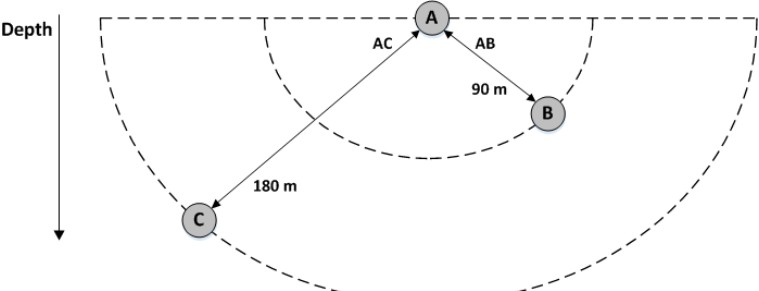

**Figure 5.** An example of the near and far problem.

### 3.5. Synchronisation

Time synchronisation is an important factor for many distributed MAC protocols and applications. It is considered to be one of the main challenges in the design of the MAC protocol, because the duty cycling approach in the MAC protocol is commonly dependent on the time synchronisation of the sensor nodes. Without accurate time synchronisation, the duty cycling cannot confirm the effective operation of the networks by handling the time uncertainty between the sensor nodes, because the propagation delay is much higher and changes over time [54,55]. The duty cycling mechanism is fully described in the following section.

### 3.6. Centralisation

There are also some issues to consider regarding the scalability of the centralised solution to a high number of density, i.e., if a base station cannot collect nodes' information, global topology, and the energy as well as the time cost of letting each node knows its schedule [31]. Therefore, a centralised network is not suitable for UWSNs over acoustic

channels, because communication between sensors nodes takes place via a central station. Moreover, due to the limited range of a single modem, the network cannot cover large areas [56].

## 4. Duty Cycle Mechanism

The duty cycle operation is widely used in sensor networks in order to improve energy efficiency [7,57,58]. It is defined as the percentage of time for which a node is active in the whole operational time. Through this principle, each node periodically switches between sleeping and listening modes. In other words, nodes are sometimes awake to either send their own data packets or receive a data packet from a neighbouring node. They are asleep during the remaining times when no data transmission or reception is occurring.

Two types of duty cycle operation can be identified: slotted listening [59,60] and, low power listening [61]. In slotted listening, as illustrated in Figure 6a, a sensor node needs to be awake during the selected slots and asleep during the remaining slots when there is no data transmission or reception. In low power listening (LPL), as depicted in Figure 6b, a sensor node needs to periodically be awake during the operational time.

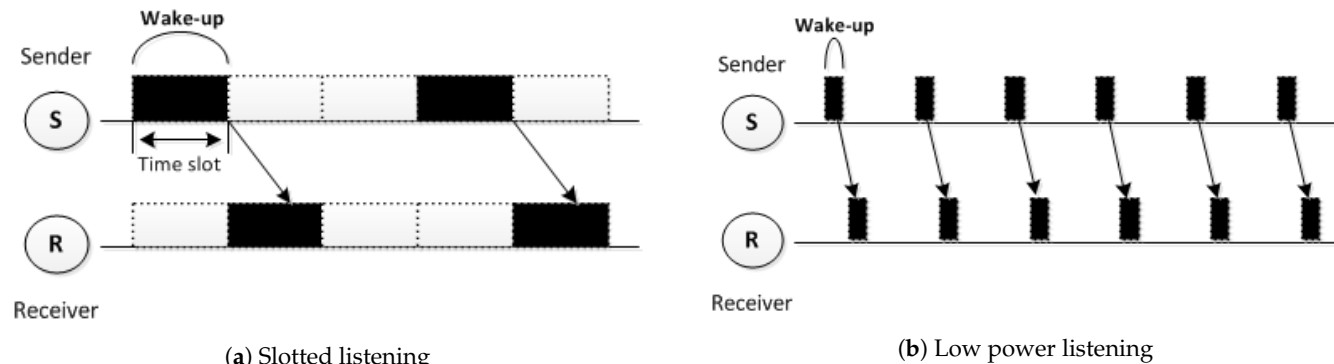

(**a**) Slotted listening          (**b**) Low power listening

**Figure 6.** Duty-cycled operation in sensor networks.

In contrast, LPL type is used for low power communication compared to slotted listing type. In other words, LPL enables radio to operate at low duty cycles. This is mainly because the duty cycle operation is commonly expressed as a percentage or a ratio. Thus, the two types of the duty cycle operation have different ratio.

## 5. Classification of Underwater Collision-Free MAC Protocols

The principle of the Medium Access Control (MAC) protocol is to manage and co-ordinate communication among nodes to access the channel. With no prior and proper management of the transmission and reception of data packets between nodes, collisions and retransmissions may occur, which will degrade the performance of the aquatic net-work. Since MAC protocols for terrestrial sensor networks use radio waves, underwater MAC protocols are significantly different as they mainly use acoustic waves as their com-munication media. The main objective of underwater MAC protocols is to consider the particular characteristics of the channel, such as high propagation delay, low data rate, and limited bandwidth, while reducing the energy cost and improving network throughput and fairness. Their design is also dependent on other factors such as scalability, reliability, and flexibility.

Due to the long propagation delay and narrow communication bandwidth in aquatic environments, the data packet transmissions consume much more energy than terrestrial networks. Because the transmission of data packets consumes more energy in UWSNs, collisions and retransmissions should be efficiently handled at the MAC layer to reduce energy consumption, and also to improve throughput and fairness across the network. However, few existing MAC protocols have attempted to reduce packet collisions and retransmissions. They have also managed to overcome conflicts between sensors and to

cope with the specific features and problems of underwater acoustic communications. A critical challenge affecting the underwater medium is time synchronisation between nodes. Since underwater applications require long-term deployment, the network may lose synchronisation over a long period of time. Hence, new medium access control techniques are needed to avoid any clock drift that may occur between nodes. Furthermore, the protocol should adopt a distributed network architecture, which allows every node to decide by itself whether to send or receive a packet, rather than one that requires centralised control, which requires a scheduler node to configure the data scheduling and pass the control packet to its neighbourhoods.

Underwater collision-free MAC protocols can typically be classified into three categories: Code Division Multiple Access (CDMA), Frequency Division Multiple Access (FDMA), and Time Division Multiple Access (TDMA) [2,16,20,33,62–66], as depicted in Figure 7. These are now described in turn.

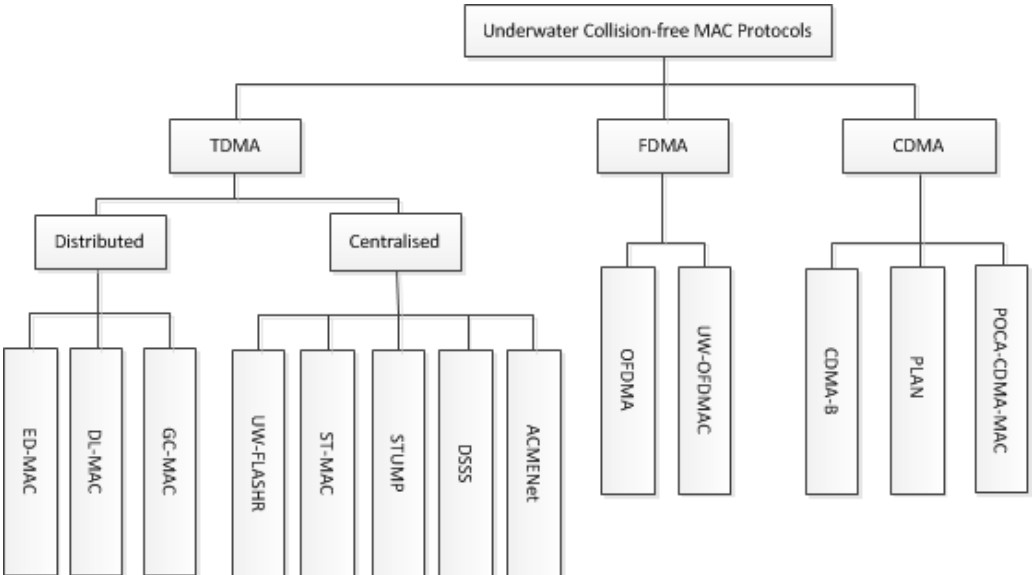

**Figure 7.** Classification of collision-free medium access control (MAC) protocols for UWSNs.

### 5.1. Code Division Multiple Access (CDMA)

The method of CDMA allows multiple sensor nodes to operate at the same time in a particular frequency band. It is robust against frequency fading, and is able to improve network throughput. Thus, destination nodes can distinguish between signals that simultaneously transmitted by several sensor nodes, which increases channel use and decreases the need for packet retransmissions. Some variances based on code-division MAC protocols have been proposed [53,67–71]. However, this method is not appropriate for UWSNs because it is difficult to set pseudo-random codes to many sensor nodes [72].

### 5.2. Frequency Division Multiple Access (FDMA)

The method of FDMA allows numbers of sensor nodes to use concurrently a specific sub-band, where the whole frequency band is divided into several sub-bands. Therefore, every sub-band is assigned by several sensor nodes to be used during the operational phase. The acoustic medium can only be used by those nodes until it is released. Several MAC protocols based on FDMA have been proposed [37,73,74]. Nevertheless, the narrow band of an acoustic channel results in a low throughput due to diffuse fading in underwater environments [56]. Hence, FDMA is not suitable and effective method for UWSNs [75,76].

### 5.3. Time Division Multiple Access (TDMA)

Using the TDMA technique, the medium channel is divided into several time slots. Every time slot can be assigned by several sensor nodes to operate at the same time. In

this multiple access technique, all sensor node are able to periodically switch among transmission, listening, and sleeping modes. This means that multiple sensor nodes can wake-up to transmit their packets or to possibly receive a packet from neighbourhood. They are in the sleeping mode during the rest of the time slots when there are no transmission and reception packets. This technique requires all the sensor nodes in the network to be remain synchronised in order to keep reliable transmission schedules, and hence no collisions can occur [77]. Therefore, TDMA is the best multiple access technique among others to be used in UWSNs due to its simplicity and flexibility. In this category, the TDMA-based protocols can further be classified into two sub-categories: centralised and distributed, which are described as follows.

### 5.3.1. Centralised

The form of centralised coordination on the network topology means a less flexible architecture. This is mainly because of a base station (i.e., a scheduler node) can control the medium access for other sensor nodes those located in its neighbourhood. MAC protocols in a centralised manner require collecting the global network's topology information, which is costly to obtain in UWSNs due to the low transmission rates and long propagation delays. Some MAC protocols based on TDMA centralised coordination have been proposed for UWSNs such as UnderWater FLASHR (UW-FLASHR) [78], Spatial-Temporal MAC (ST-MAC) protocol [2], Staggered TDMA Underwater MAC Protocol (STUMP) [8], Acoustic Communication Monitoring of Environment Network (ACMENet) [79], and Dynamic Slot Scheduling Strategy (DSSS) [80].

The UW-FLASHR protocol can achieve a higher channel use than other TDMA-based protocols. ST-MAC is also a collision-free TDMA-based protocol, which constructs a conflict graph based on global topology information. To create a conflict graph, it needs to obtain the global network's topology data, which is totally expensive to gain in UWSNs because of the low transmission rates and long propagation delays. STUMP is another typical collision-free TDMA-based protocol, where the scheduling of every sensor node is fixed for the network's entire lifetime. This strategy, however, considerably reduces channel use if the nodes' traffic loads are significantly heterogeneous. A similar approach called ACMENet which divides the sensors in the network into master and slave nodes. The master node is able to collect data packets from the slave neighbouring nodes. On the one hand, ACMENet uses the long propagation delay to avoid collisions, but on the other hand, it consumes much more energy due to idle listening. Moreover, the master node consumes higher energy than slave nodes while the lifetime of their batteries is limited which is difficult to replace or recharge. DSSS is also one of the TDMA centralised protocol which aims to increase the channel use by increasing simultaneous transmissions in parallel. Nevertheless, it requires a strict synchronisation as well as considers the transmissions of sink-to-node, node-to-node, and node-to-sink.

Overall, all these collision-free TDMA scheduling MAC protocols are typically performed in a centralised manner which is not resilient to failure [3]. Furthermore, due to the long propagation delays usually mean that a centralised MAC protocol takes a long time to collect the global topology and transmission requests from all the sensor nodes and then to notify them of the schedule, meaning that a distributed solution is preferred.

### 5.3.2. Distributed

In this sub-category, the MAC protocol should adopt a distributed network architecture, which allows every sensor node to decide individually whether to send or receive a data packet, rather than one that requires centralised control, which needs a scheduler node to configure the data scheduling and pass the control packet to its neighbourhoods. In other words, the distributed topology means there is no a scheduler sensor node to control the medium access between neighbouring nodes. Hence, all sensors are able to asynchronously deal with data transmissions and receptions. Several TDMA-based distributed collision-free MAC protocols have been proposed such as an Efficient Depth-based MAC

protocol (ED-MAC) [20,28], a Depth-based Layering MAC protocol (DL-MAC) [29,81], and a collision-free Graph Colouring MAC protocol (GC-MAC) [30,82].

In these three MAC protocols, nodes operate in three phases in which they perform asynchronously in each phase. However, they share a common clock in order to start and end every phase at the same time. Due to the fact that there is a possibility of clock drifts, a guard time interval is being used in these three protocols' algorithms to make sure that the destination nodes are able to listen prior to the source nodes start sending [83].

In terms of the performance evaluations among these three MAC protocols, an Aqua-Sim underwater simulation is used, which is an NS-2 based simulator for UWSNs [84]. Moreover, we define the most important metrics in medium access control protocols during this comparison study to evaluate the performance of ED-MAC, DL-MAC, and GC-MAC protocols as packet delivery ratio (PDR), throughput, and energy consumption per packet successfully received in joules.

Table 1 shows the requirements and properties of each of these MAC protocols along with all of their assumptions that every protocol builds on. These MAC protocols are actually classified as a TDMA-based MAC protocols, whereas they are different in terms of the required information for the operation process [85].

**Table 1.** Comparisons of TDMA-based distributed collision-free MAC protocols for UWSNs.

|  | **ED-MAC** | **DL-MAC** | **GC-MAC** |
|---|---|---|---|
| **Year** | 2017 | 2018 | 2018 |
| **Category** | TDMA-based | TDMA-based | TDMA-based |
| **TDMA status** | Adaptive slotted | Adaptive slotted | Adaptive slotted |
| **Schedule** | Distributed | Distributed | Distributed |
| **Synchronised** | Yes | Yes | Yes |
| **Clustered** | No | Yes | Yes |
| **Network division** | No | Divided into layers | Divided into cubes |
| **Priority** | Depth-based timer | Degree timer | Node ID |
| **neighbourhood info** | One-hop neighbours | One-hop neighbours | Two-hop neighbours |
| **Random time** | No | Yes | No |
| **GPS** | No | No | Yes |
| **Conflict Avoidance** | No | No | Yes |
| **Number of slots** | $2 \times N_{max}$ | Equal to $d_{max}$ | Fixed |

DL-MAC and GC-MAC protocols require network partitioning into certain layers and cubes, respectively, when being deployed, as this enhances the effectiveness of distributed network scheduling. Nonetheless, network partitions are not a requirement for ED-MAC, and its processes are not dependent on any form of clustering. On the other hand, DL-MAC and GC-MAC require these features.

During the initial stage, one-hop information needs to be collected before the scheduling stage for both ED-MAC and DL-MAC protocols. However, these protocols differ as they use different priority timers during the scheduling stage. Particularly, the timer for ED-MAC is applied to every underwater node, while reserved slots are prioritised based on the depth of the nodes. This means that a deeper node has a higher priority to reserve the first available slot time than its above neighbouring nodes during slot reservation [20,28]. For DL-MAC, sensor nodes have a degree timer, which initiates the scheduling stage. A node that has more $d$-hop adjoining nodes, and this reaches more nodes within a layer and at the range of 1-hop, and is also ranked higher priority as a potential cluster head (CH). DL-MAC's timer generates every node's degree timer while using a short and, random time interval, $\lambda$, to distinguish between sensor nodes that have similar degrees, $d_s$ [29].

In contrast, the initial stage of GC-MAC includes the collection and use of two-hop neighbouring information before the scheduling stage. The information is exchanged in this initial stage to aid with identifying any concealed terminal nodes that might not be within the two-hop neighbouring nodes. Once the sensor nodes generate their neighbouring graphs, $N_g$, the node that is closest to a pre-selected reference point, $rp$, is considered to be a CH. The CH autonomously selects its colour and assigns colours to all other one-hop neighbouring nodes independently (i.e., known as cluster members (CMs)). However, nodes placed between two cluster heads neighbours, which are located within more than two-hop neighbouring nodes, or which are not belonging to any one-hop clustering head, should select their colours separately. Therefore, node IDs with lower values are ranked higher with regard to have higher opportunity to select the first available colours between other nodes.

Based on the number of slots, these protocols have their selected algorithms and assumptions, and these are used for classifying their functioning window distinctively. For instance, ED-MAC protocol has several slots that are twice the maximum number of nodes within a neighbourhood, $N_{max}$. This aids with eliminating the potential for simultaneous information transmission between a sensor that is not within a one-hop neighbourhood and a node that is in the neighbourhood. However, DL-MAC protocol has an equal number of slots and the maximum number of nodes within a one-hop neighbourhood graphs, $d_{max}$. On the other hand, GC-MAC protocol has a certain number of slots, based on the length of the operational stage and the length of its slots. This is often equivalent to the propagation delay and the time required to ascertain that a transmission is completed before a new one starts.

Furthermore, GC-MAC is the only protocol from the above-mentioned protocols that has the phenomenon known as conflict detection (CD). The objective of CD is to identify and resolve conflicts that could occur between sensor nodes during the scheduling stage.

## 6. Qualitative Comparison

This section contains a summary of the above MAC protocols with their different scheduling designs for underwater sensor networks, as depicted in Table 2. This table includes the most commonly used collision-free MAC protocols for UWSNs, along with their sources, classifications, and a brief description of each.

Each of these collision-free MAC protocols has its own advantages and disadvantages in various scenarios. The right choice of a MAC protocol depends on the applications, network topology, network dimensions, the expected load patterns, the node deployment model (sparse versus dense sensor networks), and the specific behaviour of energy consumption. Table 3 shows the advantages and disadvantages of the above-mentioned collision-free MAC protocols. These protocols use different scheduling techniques such as scheduling-based or clustering approaches in order to avoid collisions and to conserve more energy; hence increasing the network lifetime. ED-MAC is a similar approach that uses a scheduling-based technique to achieve as many concurrent collision-free transmissions as possible in any two-hop neighbouring nodes, while DL-MAC and GC-MAC use the cluster-based technique by dividing the network into several clusters. Each cluster uses a separate transmission time to be used during the operational phase.

**Table 2.** Summary of contention-free MAC protocols with different scheduling disciplines for underwater acoustic sensor networks.

| Protocol | Source | Classification | Description |
|---|---|---|---|
| POCA-CDMA-MAC | [86] | CDMA-based | Use a round-robin method and CDMA technique to reduce data packets collisions. |
| | | Round-robin technology | Route (from sensors to the sink) is built in the first phase based on the sensor's position. |
| | | Sender-based scheduling | Every sensor follows the order to send its own packet in a round-robin manner. |
| PLAN | [70] | CDMA-based | Multiple nodes need to exchange RTS/CTS dialogue with the destination node before transmission. |
| | | Handshake-based | Exploit the ability of CDMA-based to receive concurrently from multi senders using various codes. |
| | | Receiver-based scheduling | Improve throughput while reducing packet losses arising from unsynchronised data transmissions. |
| CDMA-B | [71] | CDMA-based | Sensors are awake and sleep periodically in order to reduce undesirable energy consumption. |
| | | Multi-hop | The network is classified into multiple hierarchical levels from bottom-nodes to the top-node. |
| | | Multi-code allocation | Sensors located in same hierarchical level are multiplexed by means of various orthogonal codes. |
| UW-OFDMAC | [74] | FDMA-based | Single-hop MAC assumption. |
| | | Transmitter-based | Based on estimating the number of contenders. |
| | | OFDMA-based scheme | Random back-off promotes fairness. |
| OFDMAC | [37,73] | Reservation-based | Generates packet exchange overhead. |
| | | Handshake-based | For a dynamic topology, it is not practical to schedule transmissions |
| | | Receiver-based scheduling | at each sensor node. |
| ACMENet | [79] | Reservation-based | Provides fairness and mobility support through CSMA. |
| | | CSMA-based | Clock synchronisation is assumed. |
| | | TDMA-like approach | Sensors overhear messages to learn the transmission schedules and propagation delay for their neighbours. |
| DSSS | [80] | Reservation-based | Receiver waits for an additional duration before replying to an RTS with CTS. |
| | | Handshake-based | Accurate timing is required among sensor nodes. |
| | | Time-based | Its operation depends on the estimation of the propagation delay between two nodes. |

**Table 2.** *Cont.*

| Protocol | Source | Classification | Description |
|---|---|---|---|
| STUMP | [8] | Reservation-based | Exploit node position diversity and low propagation speed to improve the channel use. |
| | | TDMA-based | Through synchronisation, sensors share the estimated propagation delay and time slot requirements with their two-hop neighbours. |
| | | CSMA-based | Sensors are assumed to be stationary for more accuracy. |
| ST-MAC | [2] | Reservation-based | Focused on spatial-temporal uncertainty. |
| | | | Network information is collected at the sink to schedule transmissions. |
| | | TDMA-based | A centralised scheduling algorithm is used based on global topology information. |
| | | | Impairs the fairness of the network and means that some nodes starve. |
| UW-FLASHR | [78] | Reservation-based | Multiple time slots can be reserved by a single node. |
| | | Handshake-based | Nodes require an absolute time reference. |
| | | Time-slotted | A scheduled transmission cannot be cancelled even if another one is detected. |
| ED-MAC | [20,28] | Reservation-based | A packet train is formed for multiple neighbours to achieve high throughput. |
| | | Handshake-based | Sender has to know the propagation delay from itself to all intended receivers. |
| | | MACA-based | CTS packets collide when the receivers are at the same distance from the sender. |
| DL-MAC | [29] | Reservation-based | State transitions of MACA are defined according to propagation delays. |
| | | Handshake-based | Packets are assigned different priorities to avoid starvation in the case of |
| | | MACA-based | simultaneous transmission attempts. |
| GC-MAC | [30] | Reservation-based | The transmission order is determined at the receiving side. |
| | | Handshake-based | The receiver waits for RTS from all contenders. |
| | | Receiver-based scheduling | Fairness is achieved at the expense of channel use. |

**Table 3.** Comparison of different underwater collision-free MAC protocols.

| Protocol | Topology | Advantage | Disadvantage |
|---|---|---|---|
| POCA-CDMA-MAC | Multi-hop | A round-robin technique is used to receive multiple packets from neighbours simultaneously. | Multiple nodes in different paths transmit their packets periodically at the same interval time for simplicity. |
| PLAN | Distributed | Exploit the ability of CDMA-based systems to receive concurrently from multiple nodes. | Use RTS/CTS handshaking control packets which take a long time to propagate in the network. |
| CDMA-B | Multi-hop | A sleeping mode is periodically used in order to save energy when there is no data transmission or reception. | The performance of the network is highly affected by near and far problem. |
| UW-OFDMAC | Distributed | Improve energy conservation and bandwidth efficiency by adapting an orthogonal FDMA MAC framework. | The performance of UW-OFDMAC protocol is significantly affected by PAPR and ISI. |
| OFDMAC | Centralised | Adjust the orthogonal FDMA MAC framework and eliminate the hidden/exposed terminal problems. | OFDMAC's performance is affected by the diversity of frequency and multi-user. |
| ACMENet | Centralised | Exploit the high latency to avoid collisions and provide highly efficient use of the scarce network resources. | Design for a small UWSN and high energy consumption due to idle listening. |
| DSSS | Centralised | Simultaneous transmission by multi nodes increases the channel use with no chance of collision. | Accurate synchronisation is required. |
| STUMP | Centralised | Estimate the high latency to schedule conflict transmissions without requiring strict synchronisation. | Based on its scheduling constraints, some slots are not scheduled leading to insufficient channel uses. |
| ST-MAC | Multi-hop | Concentrate on spatial-temporal uncertainty and solve the conflict graph (i.e., vertex colour problem). | It is not suitable for mobile scenarios as well as the global topology information is required. |
| UW-FLASHR | Centralised | Tight clock synchronisation and accurate propagation delay are not required. | Sensors stay awake during all the established portion (i.e., second part of the cycle). |
| ED-MAC | Distributed | Guarantee collision-free scheduling while improving the energy efficiency. | Number of slots are doubled per cycle, to detect two-hop horizontal nodes, which reduces the channel uses. |
| DL-MAC | Distributed | Lower complexity using underwater features (e.g., 3D and depth of the nodes) and higher reliability and flexibility. | The scheduling packet should be forwarded up to $(d + 3)$ hops before any of them send their packets. |
| GC-MAC | Distributed | Employ the graph colouring algorithm to improve a reservation-based contention-free MAC protocol. | The location of the reference points, $rp_s$, situated in the internal cube is required (e.g., using GPS). |

## 7. Conclusions and Open Research Issues

From the aforementioned reviews, different design strategies of collision-free MAC protocols have comprehensively been studied to investigate how they handle the peculiar features of UWSNs such as long propagation delay, high mobility, limited bandwidth, and high bit error rate and how they also avoid any possibility of collisions. This is mainly because of that designing collision-free MAC protocols for underwater acoustic sensor networks is a critical task compared to that of terrestrial wireless sensor networks. However, most of the above-mentioned collision-free strategies are guaranteed to avoid collisions using scheduling-based or clustering approaches. A detailed comparison of different underwater collision-free MAC protocols with the respect to synchronisation, topology, scheduling disciplines, classification, advantages and disadvantages has been presented in Table 2. As summarised in Table 3, most of the contention-free MAC protocols use the features of the free distribution of the channel to fairly accessing the medium

and to enhance the efficiency of data packet transmission and reception as well as the energy efficiency.

The collision-free MAC protocols reviewed in this paper have widely given an overview of the current research progress on the development of the MAC protocols for the UWSNs. This survey paper shows that there is no protocol can be considered to be a perfect solution to meet all the requirements from various applications. To promote further research on the design of collision-free MAC protocols for UWSNs, we suggest the following open research issues. These are summarised below.

- A wide variety of schemes to improve underwater MAC performance have been proposed in this study. These proposals were shown to enhance the performance of the MAC protocol in different network sizes, scenarios, and applications, especially by addressing spatial-temporal uncertainty, the near-far effect, and hidden/exposed node problems. The ideas reviewed in this paper consider concepts and mechanisms that target the achievement of collision-free MAC algorithms for underwater sensor networks. Research along these lines would involve a smart MAC protocol that is able to let a sensor capture environmental characteristics with real-time predication on possible changes. This can be achieved using learning techniques, such as reinforcement learning, that solve decision problems [87–89].
- Systematic research on cross-layer designs for underwater sensor networks is vital to increase the efficiency of the MAC protocol, such as those protocols proposed particularly for cross-layer designs. They may cross either the physical or the network layers, but not both concurrently. Furthermore, they mainly concentrate on the MAC function without jointly considering other network functions for overall performance improvement, such as congestion control. A systematic design should consider all possible optimal options collectively in order to maximise performance gain [83,90].
- Finally, as the different ideas proposed in this study were based on different concepts and mechanisms of the MAC protocol for underwater sensor networks, an integration of their combined effects and algorithms should be attempted along with using the mobile Autonomous Underwater Vehicle (AUV) in a distributed manner in order to design effective AUV employed data-gathering schemes for time-critical scenarios [91,92].

**Funding:** This research received no external funding.

**Acknowledgments:** This research was conducted at the second year of career as an Assistant Professor at Naif Arab University for Security Sciences. Furthermore, I would like to express my sincere gratitude to Naif Arab University for Security Sciences (NAUSS) and the president of the university for his consistent support and encouragement.

**Conflicts of Interest:** The author declares no conflict of interest.

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
