# Peer review of "Energy-Efficient Collision Avoidance MAC Protocols for Underwater Sensor Networks: Survey and Challenges"

_jmse, doi:10.3390/jmse9070741_

Round 1

Reviewer 1 Report

This manuscript surveys the collision-avoidance multiple access control (MAC) protocols proposed in the literature for underwater sensor networks (UWSNs). It is an urgent and interesting topic from the viewpoint of readers. However, from the reference, there are only two related papers in 2020-2021. Be sure this survey covers the related papers as much as possible, especially in recent years. Besides, it might be considered to describe which kinds of performance evaluation, e.g. simulations, analysis, and experiments, etc., are applied to evaluate the performances of the proposed protocols in the literature. Moreover, it is suggested to introduce the performance metrics concerned in the MAC in the UWSNs.

Author Response

Please find attached which includes the responses to the respected reviewers. 

Thanks  

Reviewer 2 Report

The article is well organized, presenting the problems related to medium access protocols (MAC) in the case of communications in underwater sensor networks (UWSN). After commenting and detailing the typical issues, a classification of the different types of protocols that can be used to avoid communication collisions and, thus, data and energy loss is shown. Finally, a series of protocols representative of each of the above categories are analyzed and compared qualitatively.
The length of the descriptive part is much greater than that of the resolution part. A more detailed study of the different protocols selected could be carried out, providing more information on their performance.
There are some writing mistakes or incorrect expressions that could be fixed. For example, at lines 9-10, 36, 44-45, 49-50, 59-61, 110, 197, 413.
In Figure 4: "Transmission range of node C" - should be A? or the text relocated?
The "low power listening" Duty Cycle mode is not clearly explained. It would be necessary to clearly establish at what times the nodes are awake (transmitting, receiving) or asleep. In addition, the figures 6.a and 6.b do not make clear the difference between the two modes.
Most of the references are prior to 2019, which leaves out of the study some more current protocols, such as UMCI-MAC.
The article could be published with a minor revision, consisting of correcting typing errors and increasing the detail in the comparisons made with the selected protocols. 

Author Response

(The authors gave the same response as above.)
